Transmissibility of coronavirus disease 2019 in Chinese cities with different dynamics of imported cases

Chong Ka Chun 1 2
Cheng Wei 3
http://orcid.org/0000-0001-8722-6149 Zhao Shi 1 2
Ling Feng 3
Mohammad Kirran N. 1
Wang Maggie 1 2
http://orcid.org/0000-0002-7238-845X Zee Benny CY 1 2
Wei Lai 1
Xiong Xi 1
Liu Hengyan 1
Wang Jingxuan 1 jxwang@link.cuhk.edu.hk
Chen Enfu 3 enfchen@cdc.zj.cn
1 School of Public Health and Primary Care, The Chinese University of Hong Kong , Hong Kong , China
2 Shenzhen Research Institute, The Chinese University of Hong Kong , Shenzhen , China
3 Zhejiang Province Centre for Disease Control and Prevention , Hangzhou , China
Palazón-Bru Antonio
Electronic publication date: 2020 Nov 6
Publication date: 2020
Volume: 8
Electronic Location ID: e10350
Received 2020 Jun 29; Accepted 2020 Oct 21
Copyright: © 2020 Chong et al.
Copyright year: 2020
Copyright holder: Chong et al.
License: This is an open access article distributed under the terms of the Creative Commons Attribution License, which permits unrestricted use, distribution, reproduction and adaptation in any medium and for any purpose provided that it is properly attributed. For attribution, the original author(s), title, publication source (PeerJ) and either DOI or URL of the article must be cited.
License URL: https://creativecommons.org/licenses/by/4.0/

Keywords: Reproduction number, COVID-19, Serial interval, Travel restriction

Funding: National Natural Science Foundation of China 71974165 and 81473035 This work was supported by the National Natural Science Foundation of China (No. 71974165, 81473035). The funders had no role in study design, data collection and analysis, decision to publish, or preparation of the manuscript.

==============================
Background

Monitoring the reproduction number (Rt) of the disease could help determine whether there is sustained transmission in a population, but areas with similar epidemic trends could have different transmission dynamics given the risk from imported cases varied across regions. In this study, we examined the Rt of coronavirus disease 2019 (COVID-19) by taking different dynamics of imported cases into account and compared the transmissibility of COVID-19 at different intervention periods in Hangzhou and Shenzhen.

Methods

We obtained the daily aggregated counts of laboratory-confirmed imported and local cases of COVID-19 infections in Hangzhou and Shenzhen from January 1 to March 13, 2020. Daily Rt and piecewise Rt before and after Wuhan lockdown were estimated, accounting for imported cases.

Results

Since the epidemic of COVID-19 in Shenzhen was dominated by imported cases, Rt was around 0.1 to 0.7 before the Wuhan lockdown. After the lockdown of Wuhan and the initialization of measures in response to the outbreak, local transmission was well-controlled as indicated by a low estimated value of piecewise Rt, 0.15 (95% CI [0.09–0.21]). On the contrary, Rt obtained for Hangzhou ranged from 1.2 to 4.9 with a piecewise Rt of 2.55 (95% CI [2.13–2.97]) before the lockdown of Wuhan due to the surge in local cases. Because of the Wuhan lockdown and other outbreak response measures, Rt dropped below unity in mid-February.

Conclusions

Even though Shenzhen had more cases than Hangzhou, local transmission did not sustain probably due to limited transmission from imported cases owing to the reduction in local susceptibles as residents left the city during Chunyun. The lockdown measures and local outbreak responses helped reduce the local transmissibility.

Introduction

Coronaviruses are a diverse group of enveloped, positive-sense, single-stranded RNA viruses that belongs to the family Coronaviridae, order Nidovirales (Richman, Whitley & Hayden, 2016). Diseases caused by these viruses are zoonotic in nature, which can be transmitted between animals and human (World Health Organization, 2020a). In human, coronaviruses mainly cause respiratory tract infections with severe acute respiratory syndrome (SARS) and Middle East respiratory syndrome (MERS) being two notable examples. In December 2019, a novel strain of coronavirus that has not been previously identified in human emerged and caused an outbreak in Wuhan, Hubei province, China (World Health Organization, 2020b). The disease responsible for the outbreak has been officially named by the World Health Organization as coronavirus disease 2019 (COVID-19). Common clinical manifestation of COVID-19 includes fever, fatigue, dry cough, shortness of breath and muscle ache (Huang et al., 2020; Chen et al., 2020a). In severe cases, COVID-19 may progress to pneumonia or even death. Symptoms such as headache, dizziness, sputum production, hemoptysis, diarrhea, nausea and vomiting are less common but occasionally reported. According to the Centers for Disease Control and Prevention, COVID-19 is mainly spread from person-to-person via respiratory droplets, but may also be spread from contact with infected surfaces or objects (Centers for Disease Control & Prevention, 2020). At the moment, there is no vaccine nor specific antiviral treatment approved for COVID-19.

The current global scale health crisis triggered by COVID-19 began on 31 December 2019, the day the World Health Organization was notified of several severe and unusual cases of pneumonia in Wuhan. Despite the suspected source of transmission (Huanan Seafood Wholesale Market) was shut down the next day, number of cases continued to expand at an alarming rate (World Health Organization, 2020c). Health experts soon identified the cause of the disease as being a novel coronavirus that can be spread via airborne droplets and announced the discovery on 7 January 2020. Two days later, the first related death was recorded in Wuhan; meanwhile, the disease spread outside Hubei as people travel around and outside the country, especially during “Chunyun”, a 40-day period when Chinese return home for Lunar New Year reunion. On 13 January 2020, Thailand reported the first lab-confirmed case of COVID-19 outside mainland China and health authorities of 19 other countries confirmed cases over the following days (World Health Organization, 2020d). By 31 January 2020, a total of 9,826 cases of COVID-19 were confirmed around the globe and the figure surpassed 10,000 the next day.

To halt the spread of the virus, the Chinese government imposed a complete lockdown in Wuhan on 23 January 2020 aiming to quarantine the epicenter of the COVID-19 outbreak (Zhao et al., 2020a). All public transportation, including buses, railways, flights, and ferry services, was suspended and residents were forbidden from leaving the city without permission from the authorities. Later, the government tightened its quarantine measures and imposed a mandatory 14-day quarantine on people returning to the capital from holidays. Meanwhile, a number of countries have imposed restrictions on entry by travelers from China in response to the epidemic. Several early investigations have shown that the shutdown of cities largely reduced the spread of infections to other cities in China and other countries around the world (Anzai et al., 2020; Tian et al., 2020; Chinazzi et al., 2020; Zhao et al., 2020b).

The basic reproduction number (R0) is considered to be a key indicator when studying infectious disease as it reflects the ability of infection spreading, regardless the control measures taken (You et al., 2020). Estimated R0 of COVID-19 showed large variety, ranging from 0.48 in South Korea (Ki, 2020) to 6.30 in China (Sanche et al., 2020). The variation was affected by many factors including study region and period, modeling methods, targeted population and data source (Alimohamadi, Taghdir & Sepandi, 2020). According to a meta-analysis of 23 studies in China, the pooled R0 of COVID-19 was 3.32 (95% CI [2.81–3.82]) (Alimohamadi, Taghdir & Sepandi, 2020), which is much higher than that obtained in global research: Meta-analysis of 24 articles from eight countries/regions estimated a reproductive number of 2.70 (95% CI [2.21–3.30]) for COVID-19 by 10 April 2020 (Liu et al., 2020). Compared to R0, time-varying reproduction number (Rt) is a common measure for monitoring the disease evolution in a population especially when some control measures have been carried out to mitigate an epidemic. It is defined as the average number of secondary cases infected by a primary case at time t. Rt below unity indicates a single case is unable to produce more than one secondary case on average and the disease can unlikely be sustained in a population over time. In this case, the outbreak can be regarded as under control at time t (De Serres, Gay & Farrington, 2000). Publications on infectious diseases suggested that infection prevention and control measures, timely hospitalization and quarantine, sufficient healthcare resources and increased public awareness are important factors to reduce Rt, hence the risk of infectious disease transmission (Zhao et al., 2020c). During the COVID-19 pandemic, the lockdown in Wuhan was considered as an effective intervention to reduce disease transmission: median daily reproduction number decreased sharply from 2.35 on 16 January to 1.05 on 30 January (Tian et al., 2020). Province-level, weekly-based estimation also indicated a decline in Rt to <1 in Guangdong and Zhejiang between early- and mid- February (Leung et al., 2020). A more recent daily-based estimation further declared that the Rt has reduced to <1 in most provinces after 11 February (You et al., 2020). However, the effectiveness of travel restrictions varies across cities. Studies have noted that the decline in Rt was significantly associated with passenger throughput in Hubei, ratio of imported cases, control measures, and distance from Wuhan (Lau et al., 2020; Wang et al., 2020). We suggest that the effectiveness of Wuhan lockdown also varies by the dynamic of imported cases in each city. For instance, coastal cities like Hangzhou and Shenzhen had large number of migrant workers returned to their hometown during Spring Festival whom came back to these mega cities since early February. Separating the local and imported cases is essential for the estimation of policy effectiveness. In this study, we examined the transmission dynamics of COVID-19 during the first wave of epidemic and compared the transmissibility of the disease at different intervention periods in Hangzhou and Shenzhen, two Chinese cities with different transmission dynamics of imported cases.

Methods

Setting

Hangzhou and Shenzhen are two major Chinese cities in terms of economy, population and transportation. Hangzhou is the capital city of Zhejiang Province and one of the most populous cities in East China. In 2018, the city’s resident population was 9.8 million with an estimation of 4 million floating population (Municipality of Hangzhou, 2019). With 16,424 km length of highways, well-developed railway transportation and 286 civil aviation routes between 160 domestic cities, the passenger traffic of Hangzhou was over 20 million in 2018 (Hangzhou Statistics Bureau, 2018). Similar to Hangzhou, Shenzhen is also a new-growing city characterized by high-technology industry, large population of “migrant workers” and well-developed transportation network. Located in Guangdong Province, it covers an area of around 2,000 km2 and is the most populous city in South China. In 2018, the number of people living in Shenzhen was approximately 25.7 million, among whom less than half were long-term residents (Shenzhen Bureau of Statistics, 2019).

Wuhan, the source city of COVID-19, was lockdown on 23 January 2020. On the same day, Hangzhou and Shenzhen declared first-level public health emergency in response to the outbreak. Measures including promotion of personal hygiene, cessation of crowd gathering activities, 7-day home quarantine for people from Hubei, restrictions on public transports and mandatory quarantine for close contacts were adopted. On 4 February 2020, a lockdown was imposed in Hangzhou and only one person per household was allowed to leave their home every two days. Public gatherings such as funerals and weddings were banned; all public venues and workplaces were closed. On 7 February 2020, all living communities in Shenzhen were requested to lockdown in order to contain the local outbreak (Shenzhen Municipal Health Commission, 2020).

Data collection

We obtained the daily aggregated data of laboratory confirmed COVID-19 cases in Hangzhou and Shenzhen between January 1, 2020 and March 13, 2020 from the National Health Commission and other official websites (Shenzhen Municipal Health Commission, 2020; Hangzhou Center for Disease Control & Prevention (Hangzhou CDC), 2020; National Health Commission of the People’s Republic of China, 2020) (Tables S1 and S2). Date of illness onset (defined by the first appearance of COVID-19 related symptoms) was used to construct the epidemic curves. For both Shenzhen and Hangzhou, we followed the definition used in previous publication (Wu et al., 2020) and considered two types of patients as “imported cases”: (1) those who had traveled to or resided in provinces where ≥1 confirmed COVID-19 case was reported within 14 days before the onset of the disease; (2) those who had a travel history to provinces with ≥1 confirmed COVID-19 case, and did not have an obvious local source of infection.

Estimation of reproduction numbers

Since the outbreak of COVID-19 in many cities outside Hubei was seeded by imported cases, the estimation equation from Thompson et al. (2019), an alternative to Wallinga & Teunis (2004), was used, accounting for the risk of imported cases (Chong et al., 2020).

The total number of reported cases (I(t)) on day t (t = 0, 1, 2, 3…) is the sum of imported cases (hI(t)) and local cases (hL(t)). The total number of infections on day t can be computed as∑s=1twsI(t−s)=∑s=1tws[hL(t−s)+hI(t−s)]

given ws is a discretized probability distribution function of the serial interval of COVID-19. The expected number of local cases, E[hL(t)], is E[hL(t)]=Rt∑s=1tws[hL(t−s)+hI(t−s)]

Assume the number of local infections follows a Poisson distribution, a likelihood function, L(∙), is formed as follows L(Rt)=∏i=t−τt({E[hL(i)]}nL(i)e−E[hL(i)]nL(i)!)

where nL(t) is the observed numbers of local cases on day t and τ is a smoothing parameter which is assumed to be 7 days (Thompson et al., 2019). We used Markov chain Monte Carlo (MCMC) method to estimate the Rt series based on the observed epidemic curves. Using an early estimate of serial interval (Zhao et al., 2020b), we assumed the length of the serial interval follows a lognormal distribution with a mean of 4.4 days and a standard deviation (SD) of 3 days. A random walk Metropolis algorithm was used to obtain the posterior distributions of Rt. Step sizes were selected to obtain acceptance proportions of 20–40%. Twenty thousand MCMC iterations were used as the burn-in period and subsequent 100,000 iterations were used to obtain the estimates. The median and 95% credible intervals (CIs) were calculated to summarize the estimates.

In addition to the time-varying reproduction number, we estimated the piecewise Rt to compare the transmissibility of COVID-19 before and after the lockdown of Wuhan (MRC Centre for Global Infectious Disease Analysis, 2020). We assumed Rt is an exponential function of a baseline parameter (λ) and an indicator variable (X) for two periods (0-before lockdown and 1-after lockdown): Rt=λexp⁡(−αX)

where α characterises the effect of the lockdown period. Follow similar configurations (MRC Centre for Global Infectious Disease Analysis, 2020), we assumed α and λ follow a Gamma distribution (shape = 0.5, scale = 1) and a normal distribution (mean = 2, SD = 2) respectively. The posterior estimate of the piecewise Rt and the 95% CI were determined by Bayesian non-linear modeling through Hamiltonian Monte Carlo sampling. Four chains with 2,000 of warm-up and 2,000 iterations for posterior drawing were run.

The sensitivity of results to shorter serial interval of 2.0 days (SD: 2.8) (Du et al., 2020) and longer serial interval of 7.5 days (SD: 3.4) (Li et al., 2020) was tested. The analysis in this study was carried out using software R (version 3.6.3).

Results

The temporal distribution of cases and Rts in Shenzhen and Hangzhou were shown in Figs. 1A and 2A respectively. Shenzhen had the first imported case with illness onset on 1 January, whereas Hangzhou had the first imported case with illness onset on 13 January. From 1 January to 13 March, there were a total of 169 and 417 confirmed cases in Hangzhou and Shenzhen respectively. Both cities had their peak incidence of cases in the week between 22 and 29 January and the epidemics died out in mid-February.

Figure 1 Temporal distribution of cases and Rt in Shenzhen.

(A) Epidemic trend, (B) Estimate (solid line) and 95% credible intervals (dotted lines) of the time-varying reproduction number (Rt), and (C) Estimate and 95% credible band of the piecewise Rt in Shenzhen.

Figure 2 Temporal distribution of cases and Rt in Hangzhou.

(A) Epidemic trend, (B) Estimate (solid line) and 95% credible intervals (dotted lines) of the time-varying reproduction number (Rt), and (C) Estimate and 95% credible band of the piecewise Rt in Hangzhou.

In general, although Shenzhen had more cases than Hangzhou, Shenzhen had a higher percentage of imported cases than Hangzhou (83% vs 29%). Due to the epidemic in Shenzhen was dominated by imported cases, local Rt was kept below unity through time (Figs. 1B and 1C). In the early phase of the epidemic, Rt was mostly maintained at around 0.1–0.7, indicating a low risk of local transmission despite the rapid increase in daily number of cases before the lockdown of Wuhan (i.e. on 23 January). After the lockdown of Wuhan and the initialization of measures in response to the outbreak, local transmission was well-controlled as indicated by a low estimated piecewise Rt of 0.15 (95% CI [0.09–0.21]).

In contrast, Rt was larger than unity in Hangzhou from 16 January to 7 February (Figs. 2B and 2C). Before the lockdown of Wuhan, Rt obtained ranged from 1.2 to 4.9 with a piecewise Rt of 2.55 (95% CI [2.13–2.97]), indicating a high risk of local transmission. Credits to the Wuhan lockdown and other outbreak response measures, Rt dropped steadily and the corresponding piecewise Rt during this period was 0.57 (95% CI [0.31–0.82]), indicating the outbreak was controlled in Hangzhou.

Sensitivity analysis was used to assess the robustness of estimated Rt to different serial interval durations (Fig. S1). While the variation in length of serial interval did not affect the estimated Rts in Shenzhen, a moderately increased Rt was observed for Hangzhou during the first week when a longer serial interval was assumed.

Discussion

Monitoring the transmission dynamics as well as the Rt of the disease could help determine whether there is sustained community transmission in a population and evaluate whether the control measures taken are adequate to control the local transmission at a specific time (Nishiura & Cowell, 2009; Chong et al., 2017, 2018). In this study, we estimated the imported-cases-adjusted Rt of COVID-19 in the first wave of epidemic in Hangzhou and Shenzhen, two Chinese cities with different dynamics of imported cases. According to our results, the disease transmissibility in both cities gradually decreased over time especially after the lockdown of Wuhan. Credits to the community lockdown, the transmission was interrupted within a short period of time in Hangzhou. In line with Lai et al. (2020), inter-city travel restrictions and other social distancing interventions could slow down the epidemics outside Wuhan.

We showed that local transmission of COVID-19 seeded by imported cases was not sustained in Shenzhen and we speculated it was likely due to the decrease in number of local susceptibles during Chunyun. Of the 13 million residents in Shenzhen, around 65% of its population were migrants which ranked top in China, and most of the residents have returned to their hometowns starting from 10 January for celebration of Lunar New Year (Shenzhen Bureau of Statistics, 2019; Wong, 2020). Given the number of local susceptibles decreased as a large number of residents left Shenzhen during Chunyun, fewer transmission chains could be established during this period even though Shenzhen is highly connected to Wuhan for most of the times. Tian et al. (2020) indicated that Chinese cities that implemented control measures before officials confirmed the first case were more likely to have fewer cases. Even though the first case in Shenzhen was confirmed on 19 January (illness onset on 3 January), we believe Shenzhen is an exception owing to its special population characteristics. In line with Kucharski et al. (2020), the case study of Shenzhen supported that introducing one to several cases to a new city may not necessarily lead to an outbreak.

Compared to Shenzhen, Hangzhou is a highly populated city with much fewer migrants. There was an estimation of 2.3 million migrant workers in Hangzhou yet 8.2 million in Shenzhen, accounting for 28% and 65% of the city’s total population, respectively (Zeng, Yu & Zhang, 2019; Shenzhen Bureau of Statistics, 2019). The big data on Chunyun from Baidu Map (https://qianxi.baidu.com/2020/) also estimated that during January 11 and January 24, 2020, the number of people leaving Shenzhen per day is on average 3 times higher than that from Hangzhou. Geographically speaking, the distance from Hangzhou to Wuhan is approximately 336 km closer than that from Shenzhen. By implementing strict control measures before cases emerged, local transmissions were comparatively easier to be seeded in Hangzhou. Compared to the transmission dynamics in Wuhan (Kucharski et al., 2020), Rt in Hangzhou displayed a consistently declining trend staring from mid-January. This may be due to the increased awareness on the use of personal protective measures against COVID-19 after noticing the unknown pneumonia outbreak in Wuhan in early January through social media. Similar transmission dynamics were also reported in Shaanxi province (Tang et al., 2020).

In this study, we showed that even though more cases were reported in Shenzhen when compared to Hangzhou, they exhibited different transmission dynamics of COVID-19 when the risk of imported cases was taken into account during estimation. With more local cases emerged during the initial phase of the epidemic, the disease sustained in Hangzhou before the lockdown of Wuhan and the initiation of outbreak response. For cities or provinces with large proportion of imported cases, failing to differentiate imported cases from local cases during Rt calculation may lead to overestimation of transmissibility of COVID-19 in a local population (Xu et al., 2020), hence affects the planning of mitigation measures. The importance of accounting for imported cases has been demonstrated in another study that looked into MERS in Saudi Arabia (Thompson et al., 2019).

Previous scholars discussed the importance of Chunyun in COVID-19 transmission. On one hand, as migrants traveled long distance and returned to their birthplace from high-risk megacities, their travel brings pressure to disease control in their birthplaces which are usually economically and socially undeveloped regions with high proportions of elderly and less experience in epidemic management. On the other hand, outflow of migrant workers leads to decreased urban population density and lowers the risk of disease spread in migrant cities (Chen et al., 2020b; Leung et al., 2020). Although we only have data from Shenzhen and Hangzhou, a network analysis of 22 Chinese migrant cities noted the time-varying risk of COVID-19 transmission in Shenzhen is very similar to that in many other migrant cities such as Guangzhou, Beijing, Shanghai and Chongqing (Fan et al., 2020).

Our study has several major limitations. Firstly, the definition of the import cases was relied on the epidemiological investigation of travel history and there was a possibility that a case infected locally but having a travel history to other provinces 14 days prior to symptoms onset. The misclassification of the imported cases would underestimate the local transmissibility in our study. In addition, recent studies have successively reported that some patients infected with COVID-19 might infect others before their symptom onset (Du et al., 2020; Nishiura, Linton & Akhmetzhanov, 2020). Disease transmission during the pre-symptomatic stage implies the possibility of having a negative value of serial interval. This would affect the formulation of estimation equation since the distributional assumption did not give it a corresponding probability. Alternatively, generation interval could be used but it is usually hard to be estimated since the onset of infectiousness is impossible to observe. Nevertheless, as demonstrated in the sensitivity analysis, variation in serial interval within a reasonable range would not affect our main conclusion. Another limitation is the underreporting of confirmed cases due to unavailability of virological testing during the early stage of epidemic. According to an early investigation (Wu, Leung & Leung, 2020), an estimate of 75 thousand individuals were found to be infected in Wuhan as of 25 January and we believe similar underreporting was likely to occur in our setting. With data on reporting rates and serological surveillance available in detail, our analytic frame can be extended to a more complex context that incorporates these factors. In addition, our estimates shall be refined if more updated knowledge of the pathogen is available.

Conclusions

In conclusion, we showed the lockdown measures and local outbreak responses helped reduce the potential of local transmission in Hangzhou and Shenzhen. The low transmission intensity of COVID-19 in Shenzhen in early January was likely due to the decrease in number of local susceptibles as residents left the city during Chunyun. We also highlighted that, given the variation in imported cases, cities with similar epidemic trend could have different transmission dynamics.

Supplemental Information

Supplemental Information 1 Time-varying reproduction number (Rt) (left panel) and piecewise Rt (right panel) assuming a shorter (A-D) and longer serial interval (E-H) in Shenzhen (A, B, E, and F) and Hangzhou (C, D, G, and H) respectively.

Click here for additional data file.

Supplemental Information 2 Number of local and imported cases by illness onset date in Hangzhou, China.

Click here for additional data file.

Supplemental Information 3 Number of local and imported cases by illness onset date in Shenzhen, China.

Click here for additional data file.

We thank the physicians and staffs at Hangzhou, Huzhou, Jiaxing Wenzhou, Shaoxing, Ningbo, Quzhou, Jinhua, Zhoushan, Lishui, Taizhou Municipal Center for Disease Control and Prevention for their support and assistance with this investigation.

Additional Information and Declarations

Competing Interests

Author Contributions

Data Availability

MHW is a shareholder of Beth Bioinformatics Co., Ltd. BCYZ is a shareholder of Beth Bioinformatics Co., Ltd and Health View Bioanalytics Ltd. Other authors declared no competing interests.

Ka Chun Chong conceived and designed the experiments, performed the experiments, analyzed the data, prepared figures and/or tables, authored or reviewed drafts of the paper, and approved the final draft.

Wei Cheng analyzed the data, authored or reviewed drafts of the paper, and approved the final draft.

Shi Zhao analyzed the data, authored or reviewed drafts of the paper, and approved the final draft.

Feng Ling analyzed the data, authored or reviewed drafts of the paper, and approved the final draft.

Kirran N. Mohammad conceived and designed the experiments, authored or reviewed drafts of the paper, and approved the final draft.

Maggie Wang performed the experiments, authored or reviewed drafts of the paper, and approved the final draft.

Benny C.Y. Zee performed the experiments, authored or reviewed drafts of the paper, and approved the final draft.

Lai Wei analyzed the data, authored or reviewed drafts of the paper, and approved the final draft.

Xi Xiong analyzed the data, authored or reviewed drafts of the paper, and approved the final draft.

Hengyan Liu analyzed the data, authored or reviewed drafts of the paper, and approved the final draft.

Jingxuan Wang conceived and designed the experiments, performed the experiments, analyzed the data, prepared figures and/or tables, authored or reviewed drafts of the paper, and approved the final draft.

Enfu Chen conceived and designed the experiments, performed the experiments, analyzed the data, prepared figures and/or tables, authored or reviewed drafts of the paper, and approved the final draft.

The following information was supplied regarding data availability:

All the data are publicly available at the National Health Commission, Hangzhou Center for Disease Control and Prevention, and Shenzhen Municipal Health Commission.

The data are also available in the Supplemental Files.

- Hyperlink to NHC: http://www.nhc.gov.cn/wjw/yqbb/list.shtml

- to HCDCP http://www.hzcdc.net/jbkz/jbkz01.htm

- to SMHC http://wjw.sz.gov.cn

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
