# Peer review of "Transmissibility of coronavirus disease 2019 in Chinese cities with different dynamics of imported cases"

_PeerJ, doi:10.7717/peerj.10350_

## Round 0.1 · original submission · Major Revisions

Dear authors,

I have read the reviewers' reports and I think you manuscript could be suitable foe PeerJ. However, there are critical issues which you must address in a revised version of the text.

Best regards,
Dr Palazón-Bru

Reviewer 1 ·

Basic reporting

Q1: The quality of writing needs improvement, especially for grammar and typo. The following are only examples - the full text needs such review.
a. The authors name of the first page. Change “Hengyan Liu1” to “Hengyan Liu”
b. Line 191-192. Change “…the disease is useful in 1. Determining whether there is sustained community transmission in a population and 2. evaluating whether the…” to …the disease is useful in determining whether there is sustained community transmission in a population and evaluating whether the…”

Experimental design

Q2:Please include a formal case definition for “import case”. It is not clear whether the definitions in Hangzhou and Shenzhen are consistent. Besides, I note that none of the authors listed are from Shenzhen. How do you know the case definition of Shenzhen?

Validity of the findings

Q3: Your introduction needs more detail. I suggest that you improve the description at this part to provide more justification for your study (specifically, you should expand upon the knowledge R).

Q4: A short explanation on the disparity of Rs between the regions in China and other countries would be useful in the Introduction or Discussion section.

Additional comments

The article appears to lack novelty, and most of conclusions also seem hard to draw easily from the results. I suggest that authors could amplify in the Introduction and the Discussion on the added value of this study and how it is novel for this field. Besides, I don’t clearly understand the main objective of this study. However, it is also an interesting experiment to compare Rs in different regions. I suggest that you can change this article type into a Letter and resubmit it.

Reviewer 2 ·

Basic reporting

none

Experimental design

none

Validity of the findings

There was not a sufficient causation inference in the paper.

Additional comments

This article introduces us to an interesting topic. The author tried to test the hypothesis of this phenomenon with mathematical theory. But in general, there was not a sufficient causation inference in the paper. Therefore, I suggest that the author resubmit the article as a short report.
1. Line 124 to 126 .The definition of imported case was very important to the present study. The authors defined this only based on the travel history. The authors should consider the possibility that one patient was infected locally (not imported case) while he had a travel history to other provinces 14 days prior to symptoms onset. Whether the case was local or impoted should be based on the epidemiological investigation.
2. Line 201-203, 250-252. The author speculated that the low transmission intensity of COVID-19 in Shenzhen in early January was likely due to the decrease in number of local susceptibles as residents left the city during Chunyun. Was this phenomenon also occurred in other big cities such as Beijing or Shanghai? The authors should find more studies related to Chunyun to further demostrate its high influence on the transmission of COVID-19.

---

## Round 0.2 · Minor Revisions

Still pending some minor points to be considered in a new revised version of the text.

Reviewer 1 ·

Basic reporting

The writing quality of the revised manuscript has improved.

Experimental design

no comment

Validity of the findings

no comment

Additional comments

This revision is much improved after adding more background information and more references on R0. I suggest authors could provide more details (e.g.: numbers of migrant workers, the proportion of number left cities) between Shenzhen and Hangzhou.

---

## Round 0.3 · Minor Revisions

Still pending some modifications suggested by one of the reviewers.

Reviewer 1 ·

Basic reporting

The quality of writing is clear and unambiguous.

Experimental design

Methods described with sufficient detail and information to replicate.

Validity of the findings

The impact and the novelty of this study is not so great, but it is interesting to compare the Rs among different cities.

Additional comments

The quality of writing is clear and unambiguous. The impact and the novelty of this study is not so great, but it is interesting to compare the Rs among different cities. Please add the Rt of Shenzhen city in the Abstract section (results).

Reviewer 2 ·

Basic reporting

none

Experimental design

none

Validity of the findings

none

Additional comments

The author made some improvements in this revised manuscript, but did not specifically answer the previous questions.

---

## Round 0.4 · accepted · Accept

All the reviewers' concerns have been correctly addressed.